# Facile fabrication of recyclable robust noncovalent porous crystals from low-symmetry helicene derivative

Guoli Zhang[1], Jian Zhang[1], Yu Tao[2], Fuwei Gan[1], Geyu Lin[1], Juncong Liang[1], Chengshuo Shen[3] ✉, Yuebiao Zhang [2] & Huibin Qiu [1] ✉

Porous frameworks constructed via noncovalent interactions show wide potential in molecular separation and gas adsorption. However, it remains a major challenge to prepare these materials from low-symmetry molecular building blocks. Herein, we report a facile strategy to fabricate noncovalent porous crystals through modular self-assembly of a low-symmetry helicene racemate. The *P* and *M* enantiomers in the racemate first stack into right- and left-handed triangular prisms, respectively, and subsequently the two types of prisms alternatively stack together into a hexagonal network with one-dimensional channels with a diameter of 14.5 Å. Remarkably, the framework reveals high stability upon heating to 275 °C, majorly due to the abundant π-interactions between the complementarily engaged helicene building blocks. Such porous framework can be readily prepared by fast rotary evaporation, and is easy to recycle and repeatedly reform. The refined porous structure and enriched π-conjugation also favor the selective adsorption of a series of small molecules.

Porous frameworks have acquired impressive development over the past few decades and have been intensively applied in a rich variety of fields[1-4]. Conventionally, porous frameworks are constructed by covalent and coordination interactions, giving rise to covalent organic frameworks (COFs)[5-7] and metal-organic frameworks (MOFs)[8-10], respectively. Recently, noncovalent interactions including hydrogen bonds[11-15] and π-interactions[16-20] have also been utilized to fabricate porous frameworks. Due to the considerable solubility of discrete building blocks, such noncovalent porous frameworks reveal impressing solubility, processability and recyclability[18,21]. Regarding the noncovalent porous frameworks constructed majorly by π-inter-actions, the building blocks generally adopt a high-symmetry struc-ture, typically as $C_2$, $C_3$, $S_4$, etc., with relatively regular and extended ends, which subsequently stacked together to form the knot of the

framework (Fig. 1)[22-29]. So far, advances in the fabrication of porous frameworks sustained by π-interactions are substantially limited[21,30]. Besides, it remains a remarkable challenge to fabricate noncovalent porous frameworks with anomalous molecules in low symmetry.

Helicenes are a type of non-planar aromatic molecules with dis-tinctive helical π-conjugated skeletons[31,32], of which the twisted struc-tures contribute to various interesting self-assembly entities[33-36]. The stereochemical structures and widely distributed π-conjugation of helicenes benefit to the uncompact packing and the introduction of porosity[21]. Herein, we used a helicene derivative diphenyl-triphenylenyl[6]helicene (**D6H**)[37] with low symmetry to fabricate noncovalent porous frameworks merely sustained by π-interactions through modular self-assembly. Racemic **D6H** was employed and the *P* and *M* enantiomers were self-sorted upon the primary self-assembly

[1]School of Chemistry and Chemical Engineering, Zhangjiang Institute for Advanced Study, Frontiers Science Center for Transformative Molecules, State Key Laboratory of Metal Matrix Composites, Shanghai Jiao Tong University, Shanghai 200240, China. [2]Shanghai Key Laboratory of High Resolution Electron Microscopy, School of Physical Science and Technology, ShanghaiTech University, Shanghai 201210, China. [3]School of Chemistry and Chemical Engineering, Key Laboratory of Surface & Interface Science of Polymer Materials of Zhejiang Province, Zhejiang Sci-Tech University, Hangzhou 310018, China. ✉e-mail: shenchengshuo@zstu.edu.cn; hbqiu@sjtu.edu.cn

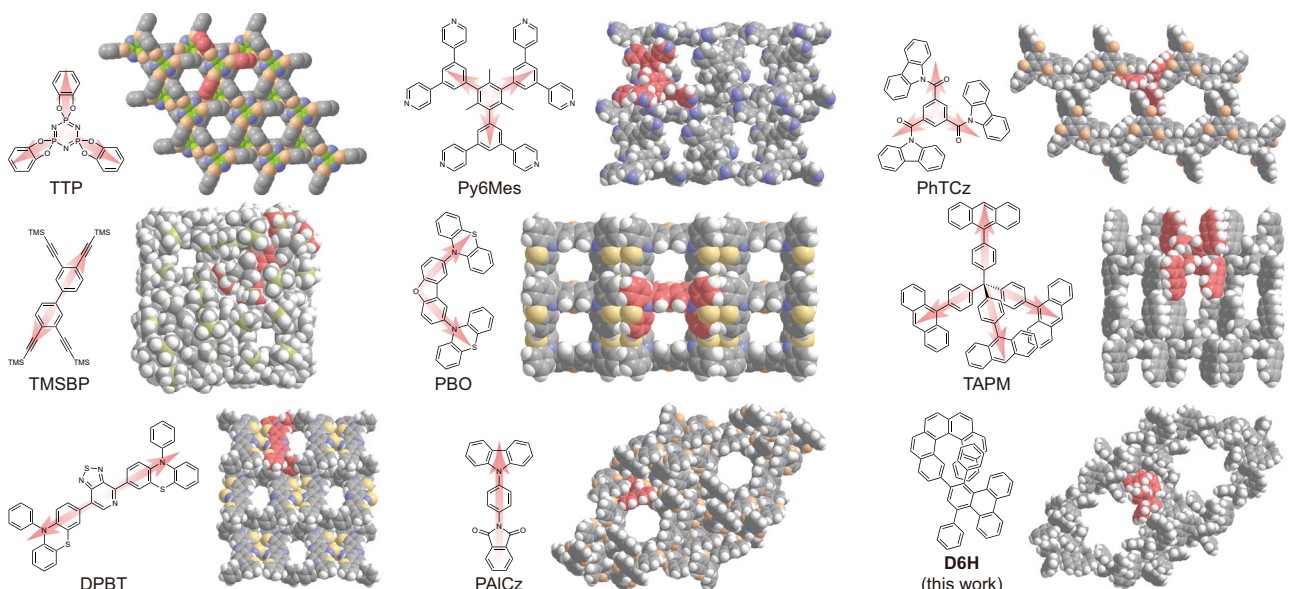

**Fig. 1 | Molecular building blocks for noncovalent porous frameworks.** Noncovalent porous frameworks formed by various organic molecules including tris-*o*-phenylenedioxycyclotriphosphazene (TTP), tris(3,5-dipyridylphenyl)mesitylene (Py6Mes), benzene-1,3,5-triyltris(9*H*-carbazol-9-yl)methanone (PhTCz), 3,3′,4,4′-tetra(trimethylsilylethynyl)biphenyl (TMSBP), 2,8-di(10*H*-phenothiazin-10-yl)dibenzofuran (PBO), tetra(9-anthracyl-*p*-phenyl)methane (TAPM), 4,7-di(10-phenyl-10*H*-phenothiazin-3-yl)[1,2,5]thiadiazolo[3,4-*c*]pyridine (DPBT), *N*-(4-(9*H*-carbazol-9-yl)phenyl)phthalimide (PAICz). The abbreviations are adopted from the literatures[22–29].

**Fig. 2 | Single crystal structure of racemic D6H. a** Space-filling representation of the crystal. Carbon atoms of *M*- and *P*-**D6H** are marked in blue and red, respectively. CH₂Cl₂ and *n*-pentane molecules possibly existing in a disordered form were removed by solvent mask using Olex2. **b** Analysis on the modular self-assembly of racemic **D6H** into a hexagonal porous framework with 1D channels.

into oppositely twisted triangular prisms, and such high-symmetry secondary building blocks with $3_1$ screw axes subsequently hexagonally stacked into a porous framework with one-dimensional (1D) channels. Crystal analysis along with calculations showed that the *P* and *M* enantiomers were associated together solely through multiple π-interactions, and thus the porous frameworks were highly stable at high temperatures, but readily recyclable/reformable through simple solvent treatment. The refined open pore structure and the pure aromatic C−H composition also favor the selective adsorption and direct analysis of various small molecules.

## Results and discussion

Single crystals of **D6H** with regular shapes were prepared by slow diffusion of a vapor of *n*-pentane into the solution of racemic **D6H** in dichloromethane (CH$_2$Cl$_2$). X-ray diffraction analysis indicated that the crystals possess a trigonal lattice in an *R*$\bar{3}$ space group (Supplementary Table 1, Supplementary Fig. 1). Interestingly, the crystal lattice showed a hollow hexagonal structure along the *c*-axis with solvent molecules randomly distributed in the channels (Fig. 2a and Supplementary Fig. 2). The hexagonally arranged 1D channels revealed a relatively large diameter of 14.5 Å (Fig. 2a). PLATON calculation[38] showed that such porous crystal has a fairly high potential volume of 32.9% for solvent encapsulation.

By further scrutinizing the crystal structure, it was found that the **D6H** molecules assembled in a hierarchical manner (Fig. 2b). First, the *P* enantiomers were self-sorted and stacked into right-handed twisted triangular prisms, while the *M* enantiomers gave left-handed prisms. In the twisted triangular prisms with $3_1$ screw axes, each helical pitch was composed of three molecules with the helicene moiety pointing inward. Despite of the irregular shape of **D6H**, the neighboring **D6H** molecules engaged with each other through multiple intermolecular C−H···π interactions with length in a range of 2.70–2.90 Å (Supplementary Fig. 3a). Subsequently, the right- and left-handed twisted

triangular prisms alternatively stacked together in a hexagonal fashion and the arch-like packing of each six prisms generated a 1D channel. Notably, the oppositely handed triangular prisms were locked with each other also through C−H···π interactions (bond lengths of 2.95 Å and 2.76 Å, respectively) (Supplementary Fig. 3b). Overall, the discrimination of chirality allowed the formation of homochiral prismatic secondary building blocks and the further generation of a heterochiral hexagonal porous framework. Interestingly, the self-assembly of the enantiopure **D6H** molecules occurred in a lamellar manner and yielded solid crystals (Supplementary Table 2, Supplementary Figs. 4 and 5), again indicating the important role of chirality discrimination in the formation of the porous crystals.

Differential scanning calorimetry (DSC) and thermogravimetric analysis (TGA) were employed to study the thermodynamic properties of the porous crystals. Upon increasing the temperature from 30 to 180 °C, a series of endothermic signals were found in the DSC curve (Fig. 3a), which could be attributed to the release of solvent molecules from the crystals. With the temperature elevated to 285 °C, a sharp endothermic peak was recorded (Fig. 3a), which may correspond to the melting of the crystals. The elimination of solvent was also reflected by the TGA curve which revealed a total weight loss of 21.8% before 285 °C (Supplementary Fig. 6). In detail, the TGA curve showed two periods of weight loss, sequentially corresponding to the detaching of surface-adsorbed solvents at lower temperatures and the escaping of encapsulated solvents upon the melting of the porous crystals. Optical microscopy observation further confirmed the melting point around 285 °C (Supplementary Fig. 7). Notably, during the heating process, the crystals kept the prismatic shape below the melting point. Variable-temperature powder X-ray diffraction (PXRD) showed that the peaks such as the ones corresponding to (220), (250), (18$\bar{1}$) and (75$\bar{1}$) faces remained constant when the temperature was increased from room temperature to 275 °C (Fig. 3b), indicating a high thermal stability of the porous crystals.

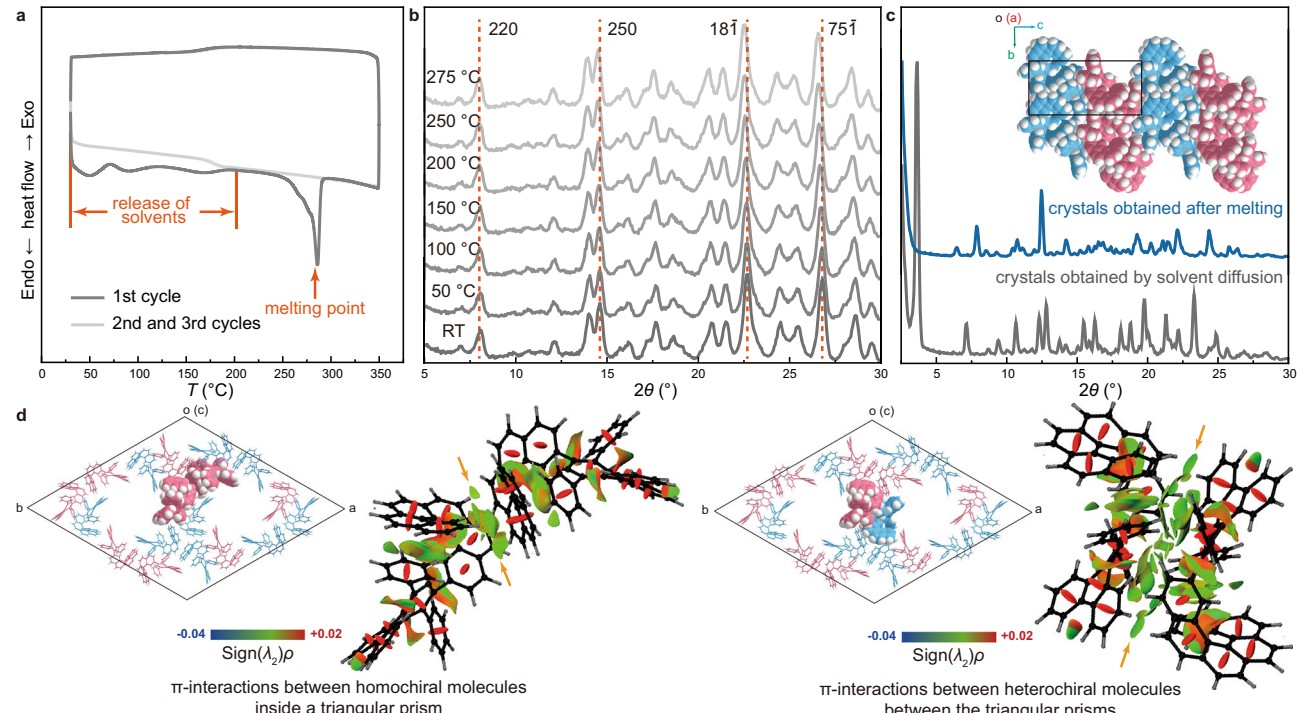

**Fig. 3 | Thermodynamic properties of D6H porous crystals. a** DSC curves of the crystals of racemic **D6H** obtained by solvent diffusion (scan rate = 10 °C/min). **b** Variable-temperature PXRD (Co Kα radiation, *λ* = 1.79021 Å) patterns of the crystals of racemic **D6H** obtained by solvent diffusion. **c** Space-filling representation of the crystals of racemic **D6H** obtained after melting and PXRD (Cu Kα radiation, *λ* = 1.54178 Å) patterns of the two types of crystals of racemic **D6H**. **d** Crystal diagrams and NCI maps showing the intermolecular interactions (denoted by arrows) between **D6H** molecules in a crystal of racemic **D6H** obtained by solvent diffusion.

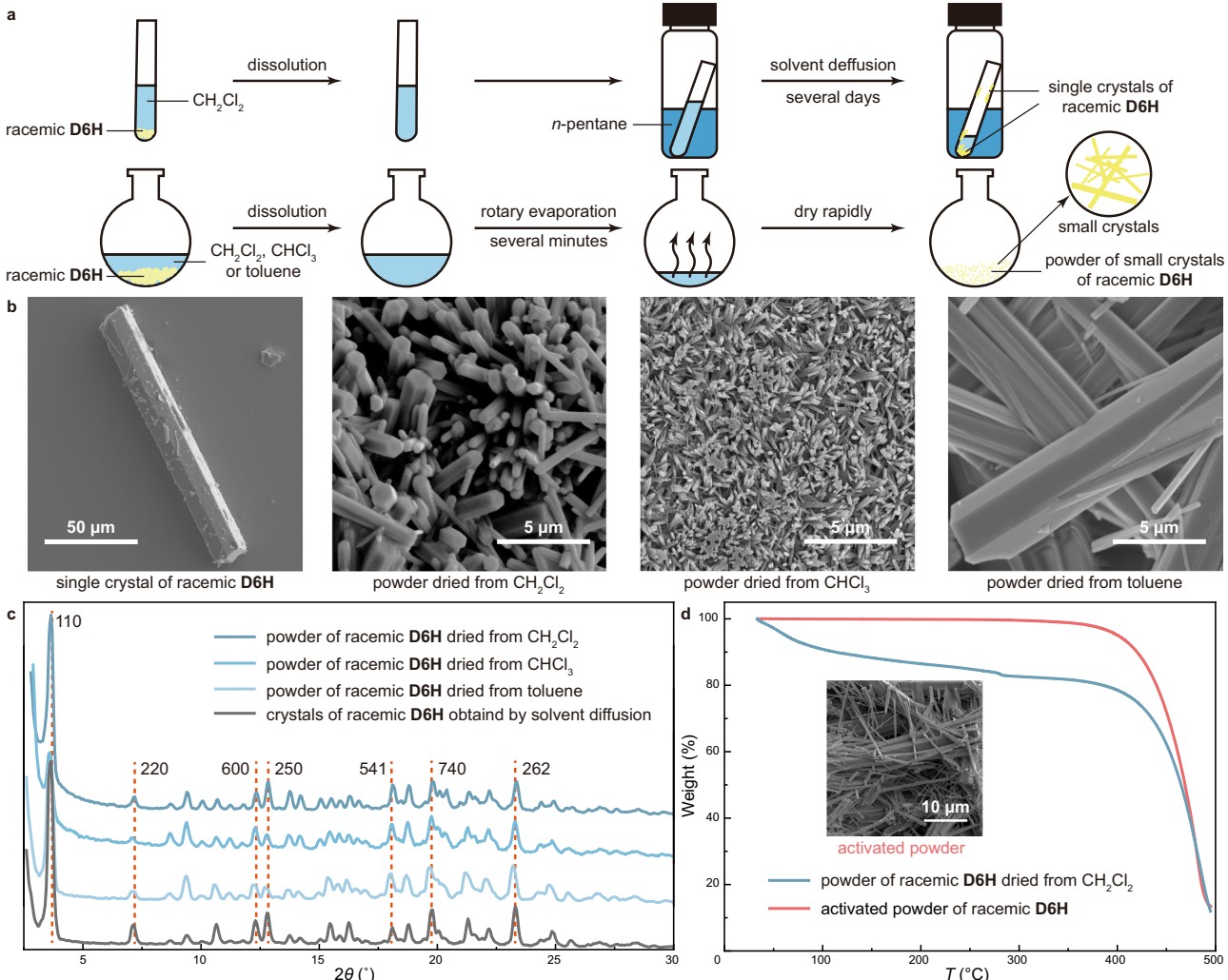

**Fig. 4 | Facile preparation of powder of D6H porous crystals. a** Schematic illustration showing the preparation of single crystals of racemic **D6H** via slow solvent diffusion and the facile preparation of powder of small crystals of racemic **D6H** via fast rotary evaporation. **b, c** Corresponding SEM images and PXRD (Cu Kα radiation, λ = 1.54178 Å) patterns. **d** TGA curves of the powder of racemic **D6H** dried from CH$_2$Cl$_2$ before and after activation (scan rate = 10 °C/min) and SEM image of the activated powder.

Interestingly, by slowly cooling the molten **D6H** from 300 to 250 °C, a new crystalline phase was formed (Fig. 3c). Single crystal X-ray diffraction analysis showed that the new crystals possess a monoclinic lattice with a $P2_1/n$ space group (Supplementary Table 3, Supplementary Fig. 8). The *P* and *M* enantiomers solely stacked into a compact nonporous lamellar structure (Fig. 3c and Supplementary Fig. 9), without any detectable residual solvents (Supplementary Fig. 10). The transition of the crystal phase was also reflected by the DSC curves (Fig. 3a and Supplementary Fig. 11). Notably, the transformation from the porous crystals to the solid crystals was thermodynamically irreversible. However, these solid crystals can be easily dissolved in CH$_2$Cl$_2$ or chloroform (CHCl$_3$) and the recycled **D6H** molecules are ready to reform the porous crystals.

Theoretical calculations were further conducted to gain more insights into the molecular association in the **D6H** porous crystals. Noncovalent interaction (NCI) analysis[39-42] revealed the presence of abundant π-interactions [−0.015 < Sign($λ_2$)$ρ$ < 0.010, in green color] between the neighboring homochiral **D6H** molecules in a single prism (Fig. 3d). The extension of triangular prisms is also accomplished by π-interactions [−0.015 < Sign($λ_2$)$ρ$ < 0.010, in green color] between the **D6H** molecules lining up along *c*-axis (Supplementary Fig. 40). On the other hand, the π-interactions [−0.020 < Sign($λ_2$)$ρ$ < 0.005, in green

color] distributed between the neighboring heterochiral **D6H** molecules assist the bundling of heterochiral triangular prisms and sustain the hexagonal porous framework (Fig. 3d). It appeared that the remarkable thermal stability of the **D6H** porous crystals originates from the abundant π-interactions[29].

Due to the lability of noncovalent interactions, noncovalent porous frameworks were normally constructed via relatively stationary crystallization approaches[21,30]. However, we found that **D6H** can spontaneously self-assemble into porous materials through fast solvent evaporation from various solvents including CH$_2$Cl$_2$, CHCl$_3$ and toluene (Fig. 4a). Scanning electron microscope (SEM) images showed that the resulting powder was comprise of micron-sized crystals with hexagonal prism morphologies similar to the single crystals of racemic **D6H** obtained by slow solvent diffusion (Fig. 4b and Supplementary Figs. 12–15). PXRD confirmed that these small crystals duplicated the highly ordered structure of the single crystals (Fig. 4c). Such swift and facilely controlled crystallization of racemic **D6H** was probably favored by the less anisotropic molecular shape which is easier to adapt a crystal lattice via fast orientation, as well as the hierarchical self-assembly manner.

The simple dissolution-evaporation process also allowed the quick and large-scale preparation of powder of small porous crystals. The

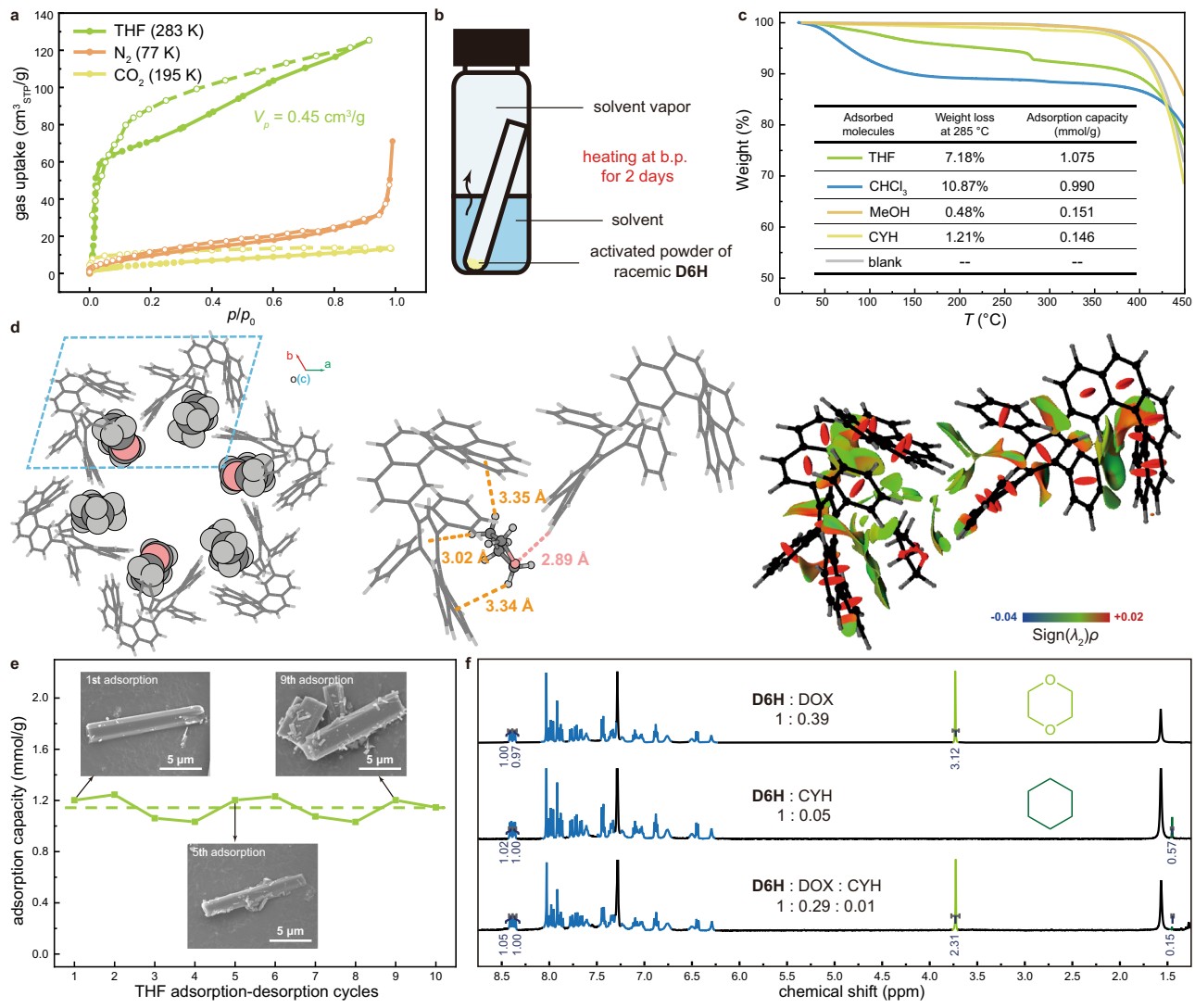

**Fig. 5 | Adsorption performance of the activated powder of racemic D6H.**
**a** Adsorption isotherms of $N_2$, $CO_2$ and THF vapor. **b** Schematic illustration of an adsorption experiment where the activated powder of racemic **D6H** is exposed to various solvent vapors for 3 days. **c** TGA curves of the activated powder of racemic **D6H** before and after the adsorption of various solvents (scan rate = 10 °C/min). The adsorption capacities are listed in the inserted table. **d** Crystal diagrams of THF@**D6H** and NCI map for the intermolecular interactions between THF and **D6H** molecules. **e** Adsorption of THF by the activated powder of racemic **D6H** upon multiple adsorption-desorption cycles. Insets are the corresponding SEM images of the powder sampled from the adsorption-desorption cycles. **f** $^1$H NMR (500 MHz, $CDCl_3$) spectra of the activated powder of racemic **D6H** after adsorption of DOX, CYH and a blend vapor of DOX with CYH.

powder of racemic **D6H** can be readily desolvated and activated by heating at 150 °C under vacuum (Fig. 4d and Supplementary Figs. 16 and 17) without any prominent interference to the crystal morphology and structure (Fig. 4d and Supplementary Figs. 18 and 19). The treated powder of racemic **D6H** can quickly take up and release iodine in an ethanol solution, indicating the activation of porosity (Supplementary Fig. 20).

The activated powder of racemic **D6H** was further subjected for gas adsorption. While the activated powder of racemic **D6H** failed to take up an adequate amount of $N_2$ or $CO_2$, it revealed a significant adsorption capacity of tetrahydrofuran (THF). The adsorption of THF vapor at 283 K revealed a pore volume ($V_p$) of 0.453 $cm^3$/g (Fig. 5a), which was close to the theoretical value (0.532 $cm^3$/g, simulated by *ZEO*++$^{43-45}$).

Subsequently, we investigated the adsorption of a series of common volatile organic solvents. The activated powder of racemic **D6H** was exposed to the vapors of a variety of solvents for 2 days to evaluate the practical gas adsorption capacity (Fig. 5b). TGA curves of the **D6H** powder collected after the adsorption of THF, diisopropyl ether (DIPE), methyl *tert*-butyl ether (MTBE), dioxane (DOX), $CHCl_3$, carbon

tetrachloride ($CCl_4$), 1,2-dichloroethane ($C_2H_4Cl_2$) and toluene revealed correspondent weight loss of 7.18%, 9.30%, 6.02%, 4.68%, 10.87%, 10.52%, 7.51% and 4.58% respectively before 285 °C, corresponding to an adsorption amount of 1.075, 1.004, 0.693, 0.566, 0.990, 0.750, 0.821 and 0.523 mmol/g under their saturated vapor pressure, respectively (Fig. 5c and Supplementary Fig. 21, Supplementary Table 4). The adsorption of THF, DIPE, MTBE, DOX, $C_2H_4Cl_2$ were also characterized by $^1$H NMR (Supplementary Figs. 22–26), and the adsorption capacity was calculated to be 1.004, 0.990, 0.622, 0.552, 0.693 mmol/g, respectively (Supplementary Table 5). On the contrary, the activated powder of racemic **D6H** showed negligible adsorption capacity towards methanol (MeOH, 0.151 mmol/g) and cyclohexane (CYH, 0.146 mmol/g) (Fig. 5c). Obviously, the adsorption feature of the activated powder of racemic **D6H** was sensitive to the polarity of small molecules$^{46,47}$. Additionally, when soaked in the MeOH solution of biphenyl, azobenzene and diphenyl disulfide respectively, the activated powder of racemic **D6H** revealed a relatively high adsorption capacity towards these aromatic derivatives (Supplementary Figs. 27–29), which demonstrated the sufficient size of the 1D

channels to accommodate larger aromatic contents. Notably, the activated powder of racemic **D6H** can be repeatedly and constantly used to capture and release the small molecules, without any prominent loss of morphological and structural integrity in 10 cycles (Fig. 5e and Supplementary Figs. 30 and 31, Supplementary Table 6).

To gain more details of the association of the porous framework with the adsorbed molecules, crystals of racemic **D6H** containing THF and CHCl$_3$ (THF@**D6H** and CHCl$_3$@**D6H**, respectively) were obtained by slow diffusion of a *n*-pentane vapor into the solution of racemic **D6H** in THF or CHCl$_3$. X-ray diffraction analysis showed that the crystals of THF@**D6H** and CHCl$_3$@**D6H** were also in an $R\bar{3}$ space group (Supplementary Tables 10 and 11, Supplementary Figs. 34 and 36). The adsorbed THF and CHCl$_3$ molecules were all located at the concave sites of the channels (Supplementary Figs. 35 and 37). In each cross section of the channel, six molecules were found to be uniformly embedded in the gap of two **D6H** molecules (Fig. 5d and Supplementary Fig. 38). In the crystal of THF@**D6H**, a THF molecule associates with a **D6H** molecule through multiple C−H⋯π interactions (bond lengths ranged from 3.00 to 3.40 Å) and simultaneously interacts with the neighboring **D6H** molecule via a C−H⋯O hydrogen bond (bond length of 2.89 Å) (Fig. 5d). Such host-guest interactions were further confirmed by NCI analysis [−0.016 <Sign($\lambda_2$)$\rho$ < 0.006, in green color] (Fig. 5d). Similarly, C−H⋯π interactions (bond length of 2.24 Å) and relative weak C−H⋯Cl hydrogen bonds (bond lengths of 3.06 Å and 3.11 Å, respectively) were found in the crystals of CHCl$_3$@**D6H** (Supplementary Fig. 39), which was also demonstrated by NCI analysis [−0.018 <Sign($\lambda_2$)$\rho$ < 0.005, in green color] (Supplementary Fig. 44).

The activated powder of racemic **D6H** was eventually applied for molecular separation. Mixtures of two target solvents in equal volume were used to generate the blend vapors for adsorption. Regarding THF and its analogues 2-methyltetrahydrofuran (MTHF), the activated powder of racemic **D6H** revealed considerably higher adsorption of THF (0.622 mmol/g) than MTHF (0.354 mmol/g) (Supplementary Fig. 32, Supplementary Table 7). For the blend of DIPE and *n*-hexane (NH), the porous powder showed an appealing adsorption selectivity of ca. 5: 1 (DIPE 0.891 mmol/g vs NH 0.184 mmol/g) (Supplementary Fig. 33, Supplementary Table 8). Notably, in spite of the greater partial pressure of CYH[48,49], the adsorption of DOX was approximately 29 times higher (DOX 0.410 mmol/g vs CYH 0.014 mmol/g) (Fig. 5f, Supplementary Table 9), indicating a remarkable discrimination between cyclic ethers and alkanes.

In summary, we have reported a distinctive type of noncovalent porous framework constructed by modular self-assembly of a low-symmetry racemic helicene derivative. The discrimination of chirality allowed the formation of homochiral triangular prism secondary building blocks and the arch-like packing of the prisms further led to the formation of a heterochiral hexagonal porous framework. The abundant π-interactions between the complementarily engaged helicene molecules favored the emergence of high thermal stability, recoverability and adsorption selectivity. Overall, this work demonstrates a facile approach to organize anomalous molecules into stable noncovalent porous crystals through hierarchical and modular self-assembly. It also broadens the utilization of π-interactions in the fabrication of noncovalent porous frameworks. The rich presence of non-planar aromatic molecules, e.g. other helicene derivatives and curved nanographenes, would pave a dynamic way for the creation of porous materials.

## Methods

### Preparation of single crystals of racemic D6H
Single crystals of racemic **D6H** were grown in CH$_2$Cl$_2$ with the diffusion of a vapor of *n*-pentane. Typically, 5.0 mg of racemic **D6H** was dissolved in 1 mL of CH$_2$Cl$_2$, and a vapor of *n*-pentane was allowed to diffuse into the solution for 3 days. The resulting crystals were washed with *n*-pentane and then dried.

### Preparation of powder of small crystals of racemic D6H
50.0 mg of racemic **D6H** was dissolved in 20 mL of CH$_2$Cl$_2$, CHCl$_3$ or toluene, respectively, and then the solvents were removed rapidly via rotary evaporation. The resulting powder was directly subjected for analysis.

## Data availability
The authors declare that all the data supporting the findings of this study are available within the article and Supplementary Information files, and also are available from the authors upon request. Crystallographic data for the structures reported in this Article have been deposited at the Cambridge Crystallographic Data Centre, under deposition numbers CCDC 2224852 (**D6H** crystallized by solvent diffusion), 2268959 (*M*-**D6H**), 2224863 (**D6H** crystallized after melting), 2255046 (CHCl$_3$@**D6H**) and 2259036 (THF@**D6H**). Copies of the data can be obtained free of charge via https://www.ccdc.cam.ac.uk/structures/. A Source Data file of the coordinates of computational optimized structures is provided with this paper. Source data are provided with this paper.

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

## Acknowledgements

This work was financially supported by the National Key R&D Program of China (2020YFA0908100, H.Q.), the National Natural Science Foundation of China (92056110, 22075180, H.Q.), the Innovation Program of Shanghai Municipal Education Commission (202101070002E00084, H.Q.), the Science and Technology Commission of Shanghai Municipality (21XD1421900, H.Q.), the Science Foundation of Zhejiang Sci-Tech University (22062026-Y, C.S.), and Zhejiang Provincial Natural Science Foundation of China under Grant (LY23B040003, C.S.).

## Author contributions

G.Z., C.S., and H.Q. conceived the project. G.Z., C.S., and F. G. synthesized the molecule and performed the X-ray crystal analysis. G.Z. collected the thermodynamic data. G.Z., Y.T. and Y.Z. conducted the sorption experiments. C.S. conducted the theoretical calculations and the data analysis. G.Z. and J.Z. performed the powder X-ray diffraction. G.Z., G.L., and J.L. acquired the SEM images. G.Z., C.S., and H.Q. analyzed the data and wrote the manuscript. All authors discussed the results, commented on the manuscript and have given approval to the final version of the manuscript.

## Competing interests

The authors declare no competing interests.
