## [Peer Review File · Nature Communications]

REVIEWERS' COMMENTS

Reviewer #1 (Remarks to the Author):

The authors have addressed all my concerns and I believe the paper is ready for publication in the current form. The work is suitable for NCOMMS and I'm happy to support publication as per my original review with the revisions made.

Reviewer #3 (Remarks to the Author):

After reviewing the authors' responses to the three reviewers, my viewpoint remains largely unchanged from my initial assessment. I still consider the scientific basis of the work to be sound, particularly with the inclusion of the experimental information that was added after the review. The concept of using helicenes as building blocks in the creation of porous crystals is interesting and synthetically noteworthy. However, to have a significant impact on a wider audience beyond Supramolecular Chemistry, there needs to be a tangible demonstration of the potential applications of this research, such as chiral recognition and optical properties. Unfortunately, the recent additions to the study do not effectively showcase the added value that helicenes could bring in this regard. As a result, I believe that the manuscript falls short of the high standards expected for publication in Nature Communications.

Response to Reviewer3

Reviewer #3 (Remarks to the Author):

After reviewing the authors' responses to the three reviewers, my viewpoint remains largely unchanged from my initial assessment. I still consider the scientific basis of the work to be sound, particularly with the inclusion of the experimental information that was added after the review.

Our reply: **We are grateful for Reviewer 3's positive comments.**

The concept of using helicenes as building blocks in the creation of porous crystals is interesting and synthetically noteworthy. However, to have a significant impact on a wider audience beyond Supramolecular Chemistry, there needs to be a tangible demonstration of the potential applications of this research, such as chiral recognition and optical properties. Unfortunately, the recent additions to the study do not effectively showcase the added value that helicenes could bring in this regard. As a result, I believe that the manuscript falls short of the high standards expected for publication in Nature Communications.

Our reply: **We thank Reviewer 3 for the thoughtful comments.**

Helicene generally possess two main features: i) structural chirality and related chiroptical characteristics, which has simulated a rich array of research works; ii) helical/curved skeleton, in terms of crystallization or self-assembly, this is extremely important. In this work, although the enantiopure helicene derivatives failed to produce porous crystals and hence the structural chirality and related chiroptical characteristics were unable to be expressed, the helical/ curved skeleton revealed an interesting role in the formation of the porous crystal. This is actually quite intriguing as it implies an innovative route for the design and fabrication of porous crystals, i.e. using helical/curved molecules. Based on this, the chiral and chiroptical properties would be additional features that can be expected.

Recently, we have synthesized several analogues of **D6H** and prepared single crystals of the racemates (Figure for Review 1). Interestingly, unlike the case of **D6H**, none of these racemic crystals were porous. The analogues revealed completely different packing modes from **D6H**. It is anticipated that the derivatives would also stack in distinct fashions compared the enantiopure **D6H**. This will probably bring new opportunities for the fabrication of chiral porous crystals.

Figure for Review 1. Analogues of **D6H** and corresponding crystal structures of racemates. Carbon atoms of *M*- and *P*-chiral molecules are marked in blue and red, respectively.